

# Estimates on age, growth, and mortality of *Leuciscus chuanchicus* (Kessler 1876) in the Ningxia section of the upper reaches of the Yellow River, China

Shuhan Xiong[1,2,3], Jiacheng Liu[1,2,3], Peilun Li[1,2,3], Yanbin Liu[4], Kai Liu[4], Yongjie Wang[4] and Jilong Wang[1,2,3]

[1] National Agricultural Experimental Station for Fishery Resources and Environment, Fuyuan, China
[2] Heilongjiang River Fisheries Research Institute, Chinese Academy of Fishery Sciences, Harbin, China
[3] Scientific Observing and Experimental Station of Fishery Resources and Environment in Heilongjiang River Basin, Ministry of Agriculture and Rural Affairs, Harbin, China
[4] Ningxia Fisheries Research Institute, Yinchuan, China

Corresponding author
Jilong Wang, wangjilong@hrfri.ac.cn

## ABSTRACT

To investigate the age structure, growth pattern, mortality and exploitation rates of *Leuciscus chuanchicus* in the upstream Ningxia section of the Yellow River, four sampling surveys were conducted between 2022 and 2023. A total of 472 individuals were measured for their total length (*TL*) and body weight (*W*). Age determination was performed using otoliths. The collected samples had a range of total lengths from 4.52 to 37.45 cm, body weights ranging from 0.68 to 552.43 g, and ages ranging from 1 to 7 years old. The relationship between total length and body weight was expressed as $W = 0.0052\,L^{3.19}$ for all samples, which indicates that the growth of *L. chuanchicus* adheres to allometry. The Von Bertalanffy growth equation revealed that the fish had an asymptotic total length ($L_\infty$) of approximately 37.9 cm with a growth coefficient (*K*) value of approximately 0.461 $yr^{-1}$. Using the age-based catch curve method, the calculated total instantaneous mortality rate (*Z*) for all samples was determined as being equal to approximately 1.1302 $yr^{-1}$. Additionally, three methods were used to estimate the average instantaneous rate of natural mortality (*M*), resulting in an approximate value of 0.7167 $yr^{-1}$ for all samples. Furthermore, the instantaneous rate of fishing mortality (*F*) for all samples was calculated as 0.4134 $yr^{-1}$, leading us to determine that the exploitation rate (*E*) is 0.3658. It was concluded that the growth rate of *L. chuanchicus* in the upstream of the Yellow River is relatively fast, and *L. chuanchicus* has not been subjected to excessive exploitation, yet its relatively high natural mortality rate underscores the need for targeted management measures aimed at preserving its habitat.

## INTRODUCTION

The upper reaches of the Yellow River serve as a crucial runoff-producing region and are characterized by abundant wildlife resources, including rare animals and plants.

Consequently, it stands as one of the key areas for wild biodiversity conservation (*Sheng et al., 2019*; *Zhao et al., 2020*). This study was conducted in the Ningxia section of the Yellow River, which traverses a fluvial plain. The region exhibits a typical continental climate characterized by arid conditions with minimal precipitation, strong winds and sandstorms, abundant sunshine, high evaporation rates, prolonged winters and short springs and summers (*He, Miao & Wen, 2013*; *Zhao et al., 2020*). Situated at an elevation ranging from 1,090 to 1,300 m above sea level (excluding the foothills of Helan Mountain where the terrain is flat and water flow is sluggish), this area encompasses numerous lakes and marshes that harbor diverse aquatic resources including rare species such as *Rhinogobio nasutus* and *Silurus lanzhouensis*. Moreover, this region serves as a national aquatic germplasm reserve in China (*Hu et al., 2022*).

In China, Mongolia, and Russia, fish species belonging to the *Leuciscus* genus fishes are economically significant and play a pivotal role in ecosystems (*Chi et al., 2010*; *Yang & Jin, 2020*). In China, it is distributed in the Yellow River and Heilongjiang basins, as well as freshwater and saline-alkali lakes in Northwest China and Xinjiang (*An et al., 2008*; *Duan et al., 2022*; *Guo et al., 2005*; *Hu et al., 2008*; *Lu et al., 2019*). The *Leuciscus* genus fishes demonstrates a strong adaptability to harsh environments, as evidenced by *Leuciscus baicalensis*'s successful introduction into a cold-water closed lake in the high mountains. Prior to this introduction, the lake had no fish species distributed within it, making it one of the few fish species capable of reproducing in this particular environment among the various introduced species (*Guo et al., 2003*).

The species *Leuciscus chuanchicus* belongs to the family *Leuciscidae*, order *Cypriniformes*, and genus *Leuciscus*. It is a pelagic fish of economic importance that inhabits cold freshwater environments characterized by slow water flow (*Xia et al., 2020*). The species is omnivorous, primarily consuming aquatic insects and copepods, while also incorporating aquatic higher plants such as diatoms and green algae into its diet (*Xia et al., 2020*). In recent years, due to the development of resources in the Yellow River, particularly illegal fishing and the unauthorized introduction of non-native fish species, there has been a 60% decline in the population of rare indigenous fish species over the past thirty years. Additionally, there has been a 50% decrease in overall fish resources. Currently, threatened fish species make up 14.7% of the total number of fish species in the Yellow River, the population status of *L. chuanchicus* is also not promising (*Zhu et al., 2022*). Therefore, conducting thorough investigation and research is imperative to safeguard the *L. chuanchicus*. Currently, the scope of research on *L. chuanchicus* is relatively limited, primarily focusing on its geographical distribution, feeding habits, sexual maturity, and other associated aspects (*Li, 2007*). Studies on age, growth, and mortality have not been reported.

The investigation of fish age, growth, and mortality is crucial for the assessment, planning, and management of fishery resources. It constitutes the fundamental aspect of studying fishery resources (*Abu El-Nasr, 2017*) Scholars frequently employ growth equations to elucidate the patterns of fish growth, and the *Von Bertalanffy (1938)* growth model is widely recognized as a precise depiction of fish growth, enjoying extensive utilization in academic research. Understanding the age structure and growth rate of fish
can reveal their life cycle, population dynamics, and population structure. This is crucial for assessing population health, ecosystem stability, and biodiversity conservation (*Rochet & Trenkel, 2003*; *Mercier et al., 2011*; *Rountrey et al., 2014*). Fish age, growth, and mortality can serve as indicators of environmental quality. By comparing fish parameters in different regions or at different time points, the impact of environmental changes on fish ecosystems can be assessed, providing important environmental monitoring data (*Santana, Dei Tos & Minte-Vera, 2020*; *Zhao et al., 2023*). Studying fish age and growth helps determine the optimal fishing time, reasonable fishing quotas, and catch limits to ensure sustainable use of fishery resources. Understanding fish mortality rates also helps evaluate the impact of fishing activities on populations and develop appropriate management measures (*Li et al., 2023*). All in all, these parameters play a significant role in fishery management.

Therefore, a sampling survey was conducted on the population of *L. chuanchicus* in the Ningxia section of the upper reaches of the Yellow River from 2022 to 2023, encompassing comprehensive examination of their individual biological characteristics. The research findings contribute to advancing our understanding of the biology of *L. chuanchicus*, fostering the implementation of effective resource management strategies, and facilitating the sustainable exploitation of this species.

## MATERIALS AND METHODS

### Sampling

The study was conducted in July 2022, March 2023, May 2023, and September 2023 using standardized net cages (length: 15 m, width: 40 cm, height: 40 cm) and gill nets (mesh sizes: 1, 2, 3 and 4 cm). A total of 472 specimens of *L. chuanchicus* were systematically captured in the Ningxia section of the upper Yellow River (Table 1). The fish samples were immediately subjected to routine biological measurements under fresh conditions including precise recordings of their total length (*TL*) up to an accuracy of ±0.01 cm and body weight (*W*) up to an accuracy of ±0.01 g. Subsequently, gender identification was performed through morphological examination based on gonadal morphology. Lapillus otoliths were carefully extracted from the inner ear sac of each specimen using tweezers followed by removal of surface connective tissue before being placed into numbered centrifuge tubes containing a solution consisting of ethanol at a concentration of 95% for preservation. Water temperature during the survey was measured using a portable water quality analyzer manufactured by HACH (Loveland, CO, USA). All sampling procedures strictly adhered to the guidelines provided by Heilongjiang River Fisheries Research Institute of CAFS for Laboratory Animal Welfare and Ethical Review.

### Otolith processing and aging

The otolith, an essential material for fish age determination, can effectively and accurately reflect the biological characteristics of fish (*Lowerrebarbieri, Chittenden & Jones, 1994*). To prepare the samples, lapilli were mounted on glass slides using transparent colorless nail polish. Subsequently, they were hand-ground with wet sandpaper (1,500 and 2,000 grit) and polished with alumina paste (3 μm) until the core and most annuli became visible under a microscope. The age of *L. chuanchicus* was determined by observing the lapillus

**Table 1 Coordinates (World Geodetic System-1984) of sampling station in Ningxia section of Yellow River.**

| Sampling station | The dates of investigation | Longitude | Latitude | The number of specimens |
|---|---|---|---|---|
| Yuding | 2022.7 | E 105.307 | N 37.472 | 21 |
| Baima | 2023.3 | E 105.545 | N 37.482 | 45 |
| Mingsha | 2023.7 | E 105.816 | N 37.563 | 53 |
| Niushoushan | 2023.9 | E 105.988 | N 37.839 | 127 |
| Yueyahu | | E 106.663 | N 38.836 | 113 |
| Taole | | E 106.831 | N 38.952 | 77 |
| Wayaocun | | E 106.773 | N 39.352 | 36 |

annuli. Furthermore, blind examination following *Li et al.*'s *(2016)* method was employed to identify the age of each otolith.

## Growth characteristics

The length–weight relationship was fitted using the power function model:

$$W = aL^b.$$

In the equation, $W$ is the body weight (g), $L$ is the total length (cm); $a$ is the condition factor for growth; and $b$ is the growth index. When $b = 3$, it indicates isometric growth, while $b \neq 3$ suggests allometry (*Cazorla & Sidorkewicj, 2008*). The $t$-test was employed to examine whether the power exponential $b$ value significantly deviated from the anticipated isometric growth three (*Ricker, 1975*). The statistical analyses were carried out using the Microsoft Excel 2019 and R 4.31.

The growth conditions are described using *Von Bertalanffy (1938)* growth equation, with the following equation:

$$L_t = L_\infty [1 - e^{-K(t-t_0)}].$$

In this equation, $t$ is age, $L_t$ is total length at age t, $L_\infty$ is asymptotic total length, $K$ is the growth coefficient, and $t_0$ represents the assumed theoretical starting age of growth. When the rate of body weight gain reaches its maximum or when the acceleration of body weight gain approaches zero, it signifies a critical inflection point in the fish's growth trajectory. The maintenance of fish resources' sustainability heavily relies on this crucial parameter. The age at which this pivotal moment occurs can be determined by employing the following formula (*Zhan, Lou & Zhong, 1986*):

$$t_p = \frac{\ln b}{k} + t_0.$$

The growth characteristic index ($\varphi$) was calculated using the formula:

$$\varphi = \lg(K) + 2\lg(L_\infty).$$

The parameters are all derived from the estimation of the growth equation (*Munro & Pauly, 1983*). Moreover, the residual sum of squares (ARSS) was employed to statistically compare the fitted growth curves between genders (*Chen, Jackson & Harvey, 1992*).

## Mortality estimation

By employing Pauly's methodology, a linear correlation was established between the natural logarithm of the sample size within each age group and their corresponding ages, resulting in an equation of the form 'y = n + mx'. In this equation, the absolute value of parameter *m* represents the total instantaneous mortality rate (*Z*) (*Beverton & Holt, 1957*; *Ricker, 1975*).

The instantaneous rate of natural mortality (*M*) is estimated more accurately by evaluating it through three empirical equations:

(1) Based on length (*Pauly, 1980*): $\ln M = -0.0066 - 0.279 \ln L_\infty + 0.6543 \ln K + 0.4634 \ln T$,
(2) Based on age (*Zhan, Lou & Zhong, 1986*; *Ralston, 1987*): $M = 0.0189 + 2.06 K$;
$$M = -0.0021 + \frac{2.5912}{t_m}.$$

The variable *T* represents the annual mean water temperature of this specific river section, while $L_\infty$ and *K* are estimated parameters for the Von Bertalanffy growth equation. Additionally, $t_m$ denotes the maximum age observed in the captured samples.

The instantaneous rate of natural mortality (*M*) refers to the relative death rate per unit time within a fish population resulting from natural factors such as predation, disease, and aging. Conversely, the instantaneous rate of fishing mortality (*F*) represents the relative death rate caused by fishing activities. These two components together constitute the total instantaneous mortality (*Z*), which can be mathematically expressed as: $Z = M + F$ (*Sainsbury & Sainsbury, 1982*).

The population exploitation rate (*E*) can be determined by dividing the fishing mortality (*F*) by the total instantaneous mortality (*Z*), as expressed in the following equation: $E = F/Z$ (*Ricker, 1975*).

# RESULTS

## Population structure

Among 472 *L. chuanchicus* samples, 240 were males and 232 were females, and the female-to-male ratio was 0.96:1 (Table 2). The total length distribution for females ranged from 4.60 to 37.45 cm, with body weight ranging from 0.68 to 552.43 g; while for males, the total length distribution ranged from 4.52 to 31.40 cm, with body weight ranging from 0.82 to 436.18 g. The majority of individuals exhibited total length concentrated between the range of 5 and 30 cm, accounting for approximately 91.94% of the population. The body weight distribution primarily ranges from 0.68 to 140 g, constituting approximately 75.00% of the total samples (Fig. 1). In a sample of 472 individuals, the mean total length was 17.96 ± 8.69 cm and the mean body weight was 97.17 ± 106.67 g.

**Table 2 Numbers of samples and total length (*L*) and body weight (*W*) in different ages of *L. chuanchicus* in the lower reaches of Yellow River.**

| Sex | Age | N | L (cm) | | W (g) | |
|---|---|---|---|---|---|---|
| | | | Min–max | Mean ± S.D | Min–max | Mean ± S.D |
| Female | 1 | 63 | 4.60–9.50 | 6.57 ± 1.09 | 0.68–6.82 | 2.28 ± 1.23 |
| | 2 | 48 | 9.30–23.3 | 16.09 ± 3.80 | 6.80–121.37 | 46.24 ± 32.60 |
| | 3 | 74 | 21.30–31.50 | 25.05 ± 2.53 | 20.35–371.88 | 144.83 ± 63.51 |
| | 4 | 29 | 22.90–35.13 | 27.69 ± 4.76 | 41.25–517.79 | 235.23 ± 112.30 |
| | 5 | 12 | 21.34–36.97 | 31.42 ± 3.57 | 103.65–526.89 | 289.12 ± 131.05 |
| | 6 | 5 | 28.50–37.00 | 32.00 ± 3.35 | 251.88–512.68 | 365.31 ± 96.00 |
| | 7 | 1 | 37.45 | 37.45 ± 0 | 552.43 | 552.43 ± 0 |
| Male | 1 | 70 | 4.52–10.72 | 6.82 ± 1.34 | 0.82–10.34 | 2.69 ± 1.75 |
| | 2 | 51 | 8.10–22.63 | 16.10 ± 3.57 | 3.04–121.21 | 43.56 ± 30.13 |
| | 3 | 76 | 12.81–27.40 | 21.33 ± 3.44 | 15.67–255.60 | 111.17 ± 48.05 |
| | 4 | 34 | 14.3–35.10 | 25.46 ± 4.20 | 60.42–418.95 | 202.91 ± 91.01 |
| | 5 | 9 | 21.35–31.40 | 25.20 ± 3.64 | 80.38–436.18 | 224.36 ± 119.37 |
| Total | 1 | 133 | 4.52–10.72 | 6.71 ± 1.23 | 0.68–10.34 | 2.49 ± 1.54 |
| | 2 | 99 | 8.10–23.30 | 16.10 ± 3.69 | 3.04–121.37 | 44.86 ± 31.39 |
| | 3 | 150 | 12.81–31.50 | 23.17 ± 3.55 | 15.67–371.88 | 130.02 ± 59.61 |
| | 4 | 63 | 14.30–35.13 | 27.22 ± 4.39 | 41.25–517.79 | 217.79 ± 102.64 |
| | 5 | 21 | 21.34–36.97 | 28.75 ± 4.74 | 60.42–526.89 | 261.37 ± 138.29 |
| | 6 | 5 | 28.50–37.00 | 32.00 ± 3.35 | 251.88–512.68 | 365.31 ± 96.00 |
| | 7 | 1 | 37.45 | 37.45 ± 0 | 552.43 | 552.43 ± 0 |

## Age structure

The otolith was used for aging of *L. chuanchicus* (Fig. 2). Through the application of alumina paste to polish the otolith and subsequent observation under an optical microscope, a discernible pattern consisting of alternating dark and bright regions can be observ (Fig. 2). The growth rings were analyzed through the observation of otolith grinding slices. The results revealed that *L. chuanchicus* individuals ranged from 1 to 7 years old (Fig. 1). Among them, age-1, age-2, and age-3 individuals were the most abundant, making up 80.73% of the total samples. Additionally, the proportion of age 4, age 5 and age 6 individuals was relatively low. Among them, there is even only one age 7 individuals.

## Growth characteristics

The relationship between total length and body weight of *L. chuanchicus* was determined through regression analysis, yielding the equation: $W = 0.0052L^{3.19}$ ($R^2 = 0.9934$) (Fig. 3). Notably, the *b* value for females was 3.24, while for males was 3.13. After conducting a *t*-test, the analysis revealed a significantly higher value of the *b* coefficient (3.19) in the relationship equation compared to 3 (*t*-test, $t = 2.97 > t_{(0.05,472)}$), indicates that the growth of *L. chuanchicus* adheres to positive allometry.
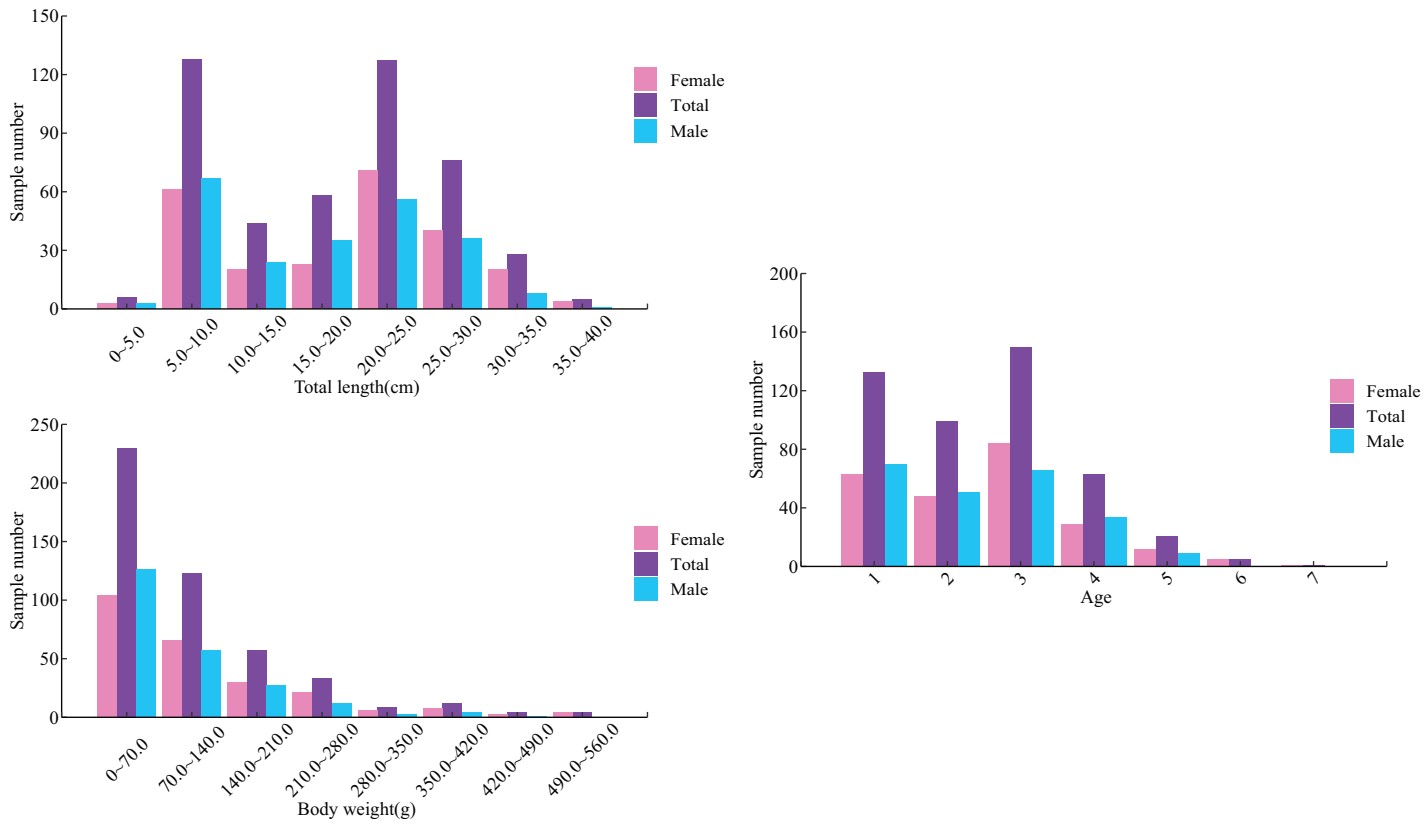

**Figure 1 The distribution of total length, body weight and age of *L. chuanchicus* in the lower reaches of the Yellow River.** Different colors represent different genders, and total represents the overall situation of both male and female.

By fitting the Von Bertalanffy equation, growth parameters for both female and male *L. chuanchicus* were estimated (Fig. 4). The growth equations for female and male are as follows:

Famale: $L_t = 39.2\left[1-e^{-0.452(t-0.597)}\right]$

Male: $L_t = 32.7\left[1-e^{-0.605(t-0.557)}\right]$.

An ARSS test was conducted to evaluate the growth disparities between males and females ($F = 0.338 < F_{(0.05,6,9)}$). The results revealed no statistically significant differences, the length growth equation for all samples was determined as

$L_t = 37.9\left[1-e^{-0.462(t-0.552)}\right]$. Moreover, the growth inflection point age has been determined as 3.068 $yr^{-1}$ using formula. Additionally, the growth characteristic index ($\varphi$) for *L. chuanchicus* is calculated as 2.820. Table 3 lists the age range, growth parameters and other indicators of several *Leuciscus* genus fishes.

## Mortality and exploitation rate

Due to the limited performance of the fishing gear in the survey, there were certain constraints encountered during data collection for age 0, age 1 and age 2. Moreover, only a single sample was available for age 7. Consequently, when computing the overall instantaneous mortality rate, data from age 1, age 2, and age 7 were excluded (Fig. 5)

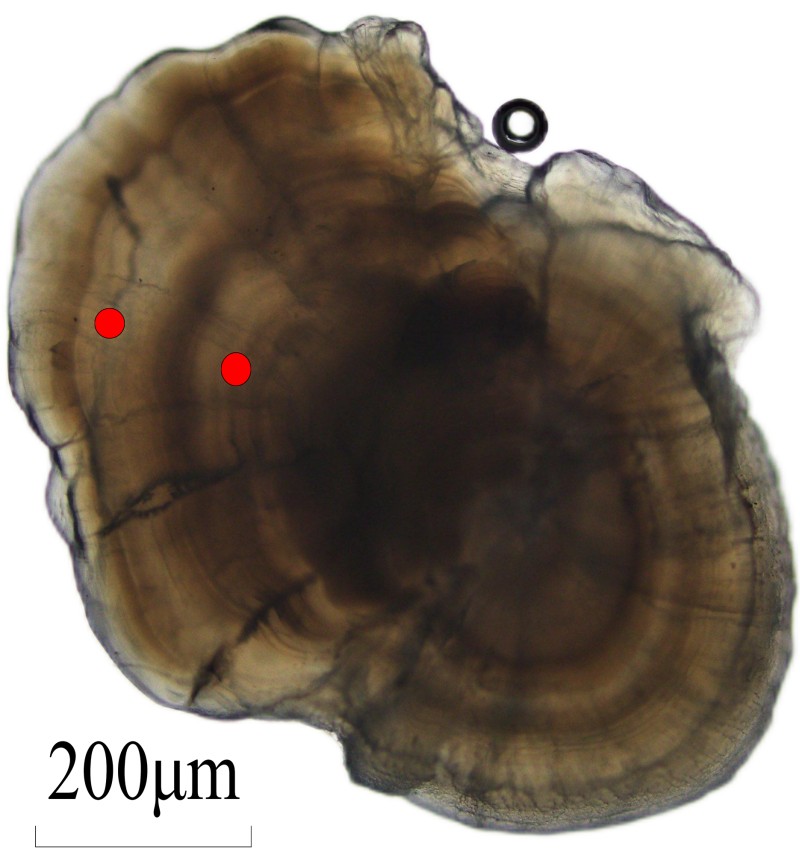

**Figure 2 Lapillus otolith section of *L. chuanchicus* (*L* = 26.80 cm, *W* = 157.02 g, age-3).** The red circle indicate the otolith ring pattern consists of two light and dark rings, so the age of the sample is 2+ years old (age-3).                                               

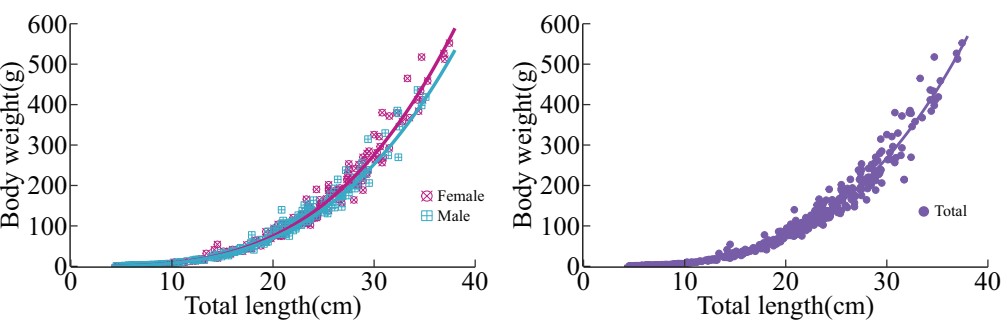

**Figure 3 Length-weight relationships for females, males and total of *L. chuanchicus* in the lower reaches of the Yellow River.** The left graph shows the relationship between length and weight separated by gender, while the right graph represents the overall relationship between length and weight in the sample.                                               

(*Beverton & Holt, 1957*). The total instantaneous mortality rate (*Z*) was determined to be 1.1302 $yr^{-1}$.

The estimation of the instantaneous rate of natural mortality (*M*) for the entire sample is calculated using three empirical formulas (*Pauly, 1980*; *Zhan, Lou & Zhong, 1986*; *Ralston, 1987*), resulting in a value of 0.7167 $yr^{-1}$. Subsequently, employing their

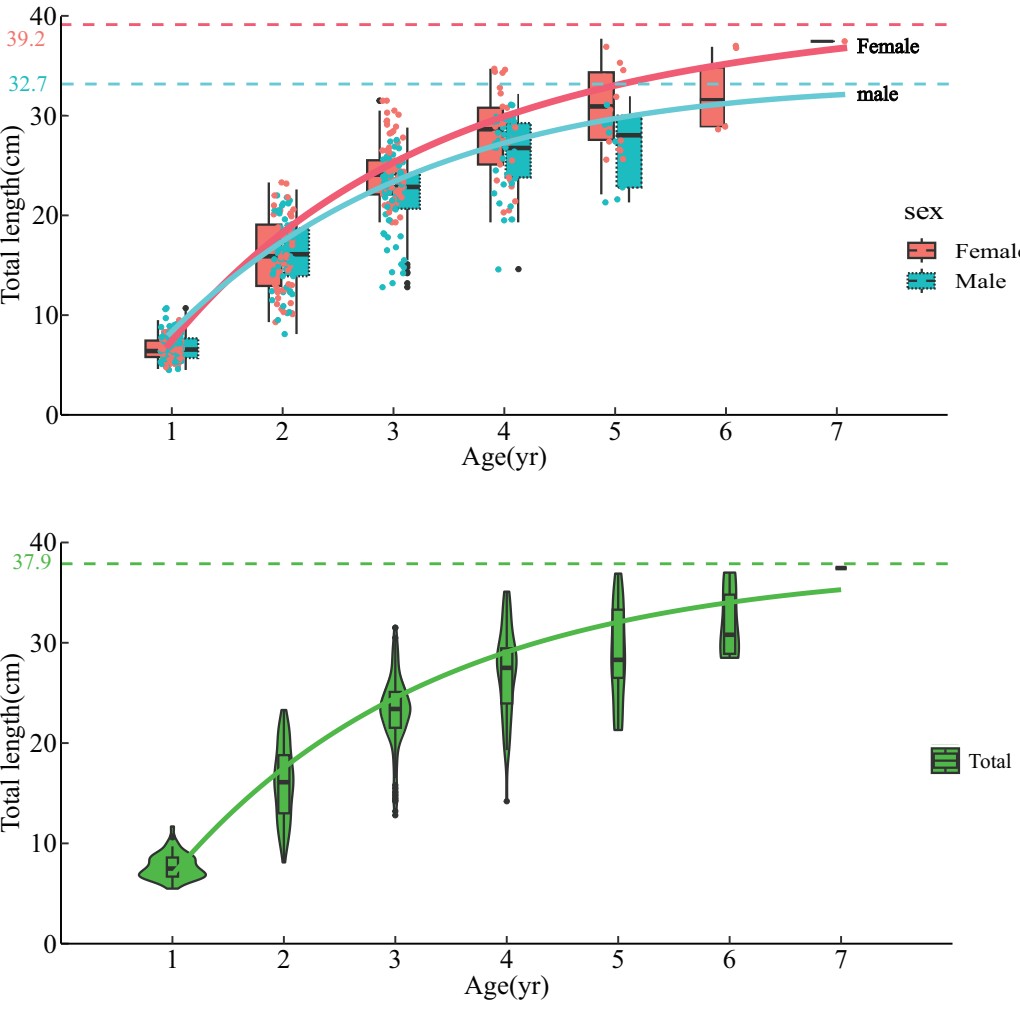

**Figure 4 Von Bertalanffy growth formula fitted to females, males and total length-at-age for *L. chuanchicus* from samples captured.** The dashed line represents the asymptotic total length. The dots represent the distribution of total length corresponding to each age group, and the box plot and violin plot provide a visually clear representation of these data points. The dashed line represents the overall asymptotic total length.

mathematical correlation, the corresponding instantaneous rate of fishing mortality ($F$) is computed as $0.4134 \ yr^{-1}$. Finally, based on these values, an exploitation rate ($E$) of 0.3658 is determined. Furthermore, pertinent parameters for both females and males were separately determined as presented in Table 4.

## DISCUSSION

### Age and growth

Although many scholars usually use scales as *Leuciscus* genus fishes age identification materials (*Hu et al., 2008*; *Qi et al., 2011*), scales can be difficult to read and may have incomplete growth history due to loss of scales, leading to inaccurate age estimation (*Campana, 2001*; *Howland et al., 2004*). Otoliths serve as a reliable method for determining fish age, characterized by the formation of distinct growth rings and maintenance of a

**Table 3 Comparison of growth characters of *Leuciscus* fishes in different studies.**

| Species | Sex | Age range | Growth parameters | | | Sources |
|---|---|---|---|---|---|---|
| | | | $L_\infty$ (cm) | $K$ | $\varphi$ | |
| *Leuciscus merzbacheri* | ♀ and ♂ | 1–4 | 24.27 | 0.3217 | 2.277 | *Guo et al. (2005)*, *Hu et al. (2008)* |
| *Leuciscus idus* | ♀ and ♂ | 1–5 | 38.28 | 0.3269 | 2.680 | *Hu et al. (2008)* |
| *Leuciscus baicalensis* | ♀ and ♂ | 1–3 | 32.96 | 0.2402 | 2.416 | *Hu et al. (2008)* |
| *Leucisus waleckii* | ♀ and ♂ | 1–6 | 35.75 | 0.26 | 2.521 | *Lu et al. (2019)* |
| *Leucisus waleckii* | ♀ | 1–7 | 49.73 | 0.199 | 2.692 | *Duan et al. (2022)* |
| | ♂ | 1–6 | 41.59 | 0.27 | 2.669 | |
| *Leucisus waleckii* | ♀ and ♂ | 2–8 | 28.97 | 0.2586 | 2.336 | *An et al. (2008)* |
| *Leuciscus chuanchicus* | ♀ and ♂ | 1–7 | 38.3 | 0.501 | 2.866 | This study |

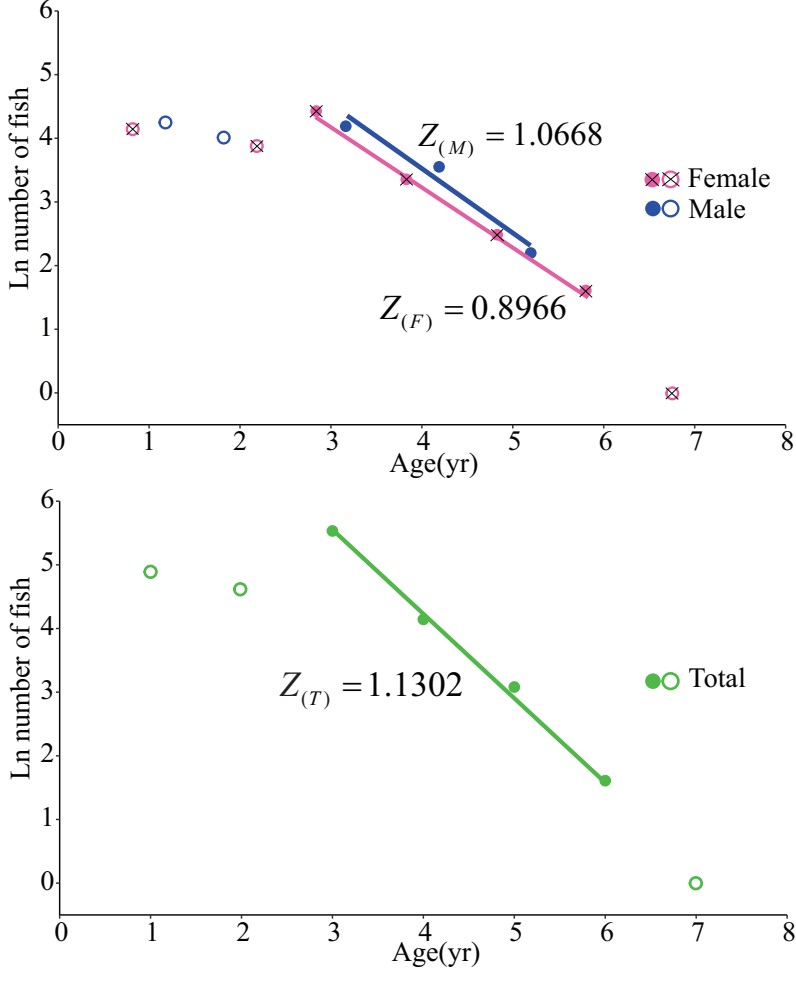

**Figure 5 Catch curve based on observed age for *L. chuanchicus* samples.** Z represents the total instantaneous mortality rate; Empty points are the data to be excluded.

**Table 4 Parameters related to mortality rates in females, males and total.**

| Sex | Z | M | F | E |
|---|---|---|---|---|
| Female | 0.8966 | 0.7045 | 0.1920 | 0.2141 |
| Male | 1.0668 | 0.8819 | 0.1848 | 0.1732 |
| Total | 1.1302 | 0.7167 | 0.4134 | 0.3658 |

stable chemical composition throughout their development. Despite eventual growth cessation, the recorded information remains preserved without any reabsorption (*Campana & Thorrold, 2001*). In cyprinids, the lapillus is the largest otolith and therefore preferred over the sagitta for fish aging purposes (*Rohtla et al., 2015*). It should be noted that otoliths require sacrificing the fish which may pose a problem when dealing with endangered species or populations (*Howland et al., 2004*; *Zymonas & McMahon, 2009*).

The age distribution of the *L. chuanchicus* population in the investigated area ranges from 1 to 7 years, with a relatively lower proportion of individuals aged 5, 6, and 7. The age structure exhibits a relatively simplified pattern. The age structure of *L. chuanchicus* is higher compared to that of *L. merzbacheri* (1–4) (*Guo et al., 2005*; *Hu et al., 2008*), *L. idus* (1–5) (*Hu et al., 2008*), and *L. baicalensis* (1–3) (*Hu et al., 2008*) captured within China. It exhibits similarities with *L. waleckii* (1–7) (2–8) (*An et al., 2008*; *Lu et al., 2019*; *Duan et al., 2022*), but significantly lower than the age structure observed in *Leuciscus idus* (1–29) (*Rohtla et al., 2015*) caught in the Baltic Sea.

The *b* value represents the relationship between total length and body weight, which not only varies among different species but also differs among individuals of the same species or population, as well as at different developmental stages, reflecting variations in stomach fullness, overall appetite condition, and gonadal stages (*Flura et al., 2015*). In this study, the *b* value of *L. chuanchicus* is 3.17 in winter, whereas it is 3.18 in summer, exhibiting minimal disparity between the two seasons, the overall *b* value of the *L. chuanchicus* population is 3.19, indicating allometry. It is comparable to *L. waleckii* (3.17; 3.18), higher than *L. baicalensis* (2.84) and *L. idus* (2.97), but lower than *L. merzbacheri* (3.29). Additionally, the *b* value of females (3.24) surpasses that of males (3.13), suggesting a greater inclination among females to prioritize weight gain as a means to enhance reproductive investment. This adaptive strategy reflects the population's response to its environment (*Zhao et al., 2023*).

The *K* value is commonly employed to indicate the rate of population growth. *Branstetter (1987)* categorized the growth coefficient into three groups: 0.05–0.1 represents species with slow growth; 0.1–0.2 represents species with moderate growth; 0.2–0.5 represents species with rapid growth. The *K* value of *L. chuanchicus* is 0.461, higher than the documented *K* values (0.1990–0.3269) (*An et al., 2008*; *Duan et al., 2022*; *Guo et al., 2005*; *Hu et al., 2008*; *Lu et al., 2019*) for the *Leuciscus* genus fishes. The asymptotic total length $L_\infty$ of females (39.2 cm) exceeds that of males (32.7 cm), while their *K* value is comparatively smaller. Consequently, females require more time to attain their asymptotic total length in comparison to males. This indirectly reflects the prioritization of

reproductive capacity and successful reproduction by females, potentially resulting in slower growth rates. Conversely, males are inclined to allocate resources towards growth and competition, leading to accelerated growth rates (*Luo et al., 2017*). In addition to habitat factors, the size range of captured samples also influences the *K* value estimation. Therefore, it is crucial to adjust fishing gear and select appropriate fishing locations for obtaining precise parameter estimates (*Campana, 2001*).

The growth characteristic index ($\varphi$) can be utilized for intra-genus comparison of fish growth performance (*Munro & Pauly, 1983*; *Pauly, 1987*). The growth characteristic index of *L. chuanchicus* in this study is 2.820, surpassing the range observed in all known *Leuciscus* genus fishes (2.277–2.692) (*An et al., 2008*; *Duan et al., 2022*; *Guo et al., 2005*; *Hu et al., 2008*; *Lu et al., 2019*). Consequently, the *L. chuanchicus* is classified as a species exhibiting excellent growth performance among the *Leuciscus* genus fishes. The growth inflection point age (3.068 $yr^{-1}$) for *L. chuanchicus* was subsequently determined using a mathematical formula. The proportion of individuals at the growth inflection point age (3 $yr^{-1}$) in the total catch is only 31.77%, while individuals at age-1 and 2 constitute a larger portion of the overall population (49.15%). This finding suggests that current fishing practices may exert an adverse impact on the population recovery of *L. chuanchicus* (*Ye et al., 2012*).

## The mortality and exploitation rate of *L. chuanchicus*

The exploitation rate (*E*) of fish is a crucial parameter in fisheries management (*Yao et al., 2018*). It is imperative to maintain the exploitation intensity for fish below 0.5, as surpassing this threshold would result in overexploitation and have detrimental effects on the sustainability of fish stocks (*Hu et al., 2012*; *Sun et al., 2021*). In this study, three empirical formulas were employed to calculate the instantaneous natural mortality rate (*M*). As a result, an exploitation rate (*E*) of 0.3658 was obtained for *L. chuanchicus*, which remained within the acceptable range of exploitation intensity (<0.5). The instantaneous natural mortality rate of *L. chuanchicus*, however, is relatively high; females exhibit a higher natural mortality rate compared to males. The final results reveal that the influence of human fishing activities on the *L. chuanchicus* is relatively insignificant, with a greater contribution stemming from environmental pressures. The section of the river under discussion is situated within a national aquatic germplasm reserve in China (*Hu et al., 2022*), where local authorities enforce stringent regulations to combat illegal fishing activities. However, it is important to note that this particular area also serves as a crucial Yellow River irrigation district, providing substantial benefits to a population of approximately 5.6 million individuals (*Gao et al., 2023*). In the summer, the visibility in this section of the river can be reduced to less than 1 cm, exerting a significant impact on aquatic organisms (*Niu et al., 2024*), where inadequate regulation has resulted in the escape of farmed species, leading to the establishment of invasive alien species (*Vythalingam et al., 2022*). The aforementioned factors have contributed to a relatively elevated natural mortality rate of *L. chuanchicus*. To address this issue, we propose the implementation of the following measures: (1) Efforts to combat illegal fishing and enforce stringent penalties should be strengthened in order to enhance law enforcement. Moreover, the promotion of

public awareness and engagement in safeguarding fish resources can be achieved through educational initiatives and outreach campaigns. (2) Monitoring of Yellow River water quality should be enhanced, including measures such as reducing industrial wastewater discharge and controlling agricultural pollutants. In particular, intensified efforts should be made to invest in tributaries management with the aim of improving the water quality of tributaries that flow into the Yellow River. (3) Strengthen the management of alien species, especially in terms of preventing escape and private release control of farmed fish, promptly detect and address issues related to invasive alien species.

## CONCLUSION

The findings suggest that *L. chuanchicus* exhibits positive allometry and excellent growth performance ($\varphi$ = 2.820), with an asymptotic total length ($L_\infty$) of 37.9 cm, when compared to other species within the same genus. This species demonstrates a relatively rapid growth rate, characterized by a $K$ value of 0.461. Furthermore, the study indicates that the population of *L. chuanchicus* has not experienced excessive fishing pressure; however, it faces high natural mortality rates. Therefore, local implementation of management strategies such as water quality monitoring and optimization, as well as control measures against non-native fish introductions are crucial.

### Funding

This work was funded by the Project of Yellow River Fisheries Resources and Environment Investigation from the MARA, P. R. China: HHDC-2023-02; the investigation on alien invasive aquatic animals in the Yellow River: ZF2022512304; and the China Academy of Fishery Sciences: 2023TD07. The funders had no role in study design, data collection and analysis, decision to publish, or preparation of the manuscript.

### Grant Disclosures

The following grant information was disclosed by the authors:
MARA, P. R. China: HHDC-2023-02.
Investigation on Alien Invasive Aquatic Animals in the Yellow River: ZF2022512304.
China Academy of Fishery Sciences: 2023TD07.

### Competing Interests

The authors declare that they have no competing interests.

### Author Contributions

- Shuhan Xiong conceived and designed the experiments, performed the experiments, analyzed the data, prepared figures and/or tables, authored or reviewed drafts of the article, and approved the final draft.
- Jiacheng Liu performed the experiments, authored or reviewed drafts of the article, and approved the final draft.
- Peilun Li analyzed the data, prepared figures and/or tables, and approved the final draft.

- Yanbin Liu performed the experiments, prepared figures and/or tables, and approved the final draft.
- Kai Liu performed the experiments, prepared figures and/or tables, and approved the final draft.
- Yongjie Wang analyzed the data, prepared figures and/or tables, and approved the final draft.
- Jilong Wang conceived and designed the experiments, authored or reviewed drafts of the article, and approved the final draft.

### Animal Ethics

The following information was supplied relating to ethical approvals (*i.e.*, approving body and any reference numbers):

The Experimental Animal Welfare Ethics Committee of the Heilongjiang Fisheries Research Institute provided full approval for this study (2021.12.10).

### Data Availability

The raw data is available in the Supplemental File.

### Supplemental Information

Supplemental information for this article can be found online at http://dx.doi.org/10.7717/peerj.17351#supplemental-information.

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
