# Peer review of "Estimates on age, growth, and mortality of Leuciscus chuanchicus (Kessler 1876) in the Ningxia section of the upper reaches of the Yellow River, China"

_PeerJ, doi:10.7717/peerj.17351_

## Round 0.1 · original submission · Minor Revisions

Dear Dr. Xiong

The reviewers have commented on your manuscript. Based on the comments and suggestions of the expert reviewers, a minor revision is needed for your article.

I would like to request that you check and correct the manuscript based on the reports.

**Language Note:** The Academic Editor has identified that the English language must be improved. PeerJ can provide language editing services - please contact us at [email protected] for pricing (be sure to provide your manuscript number and title). Alternatively, you should make your own arrangements to improve the language quality and provide details in your response letter. – PeerJ Staff

Sincerely yours

Reviewer 1 ·

Basic reporting

There are many grammatical errors. The language should be checked.

No professional article structure

Experimental design

It is important as it is the first detailed study on the species, but unfortunately, there are erroneous calculations.
First of all, the current position of Leuciscus chuanchicus in the systematics should be checked.
The asymptotic length calculated for males was lower than that of the biggest individual captured, indicating an error in the calculations. Therefore, it is necessary to re-determinate the age and then the calculations.
The fact that individuals belonging to 0 age group were not captured may be due to net selectivity.

Validity of the findings

It is important as it is the first detailed study on the species, but unfortunately, there are erroneous calculations.
Conclusion is poor.

Additional comments

All comments are included in the attached file.

Annotated reviews are not available for download in order to protect the identity of reviewers who chose to remain anonymous.

Reviewer 2 ·

Basic reporting

I recommend a check by a fluent speaker due to some general minor issues with the English language.

Authors are interchanging the passive and active voices throughout the manuscript, it is not necessarily wrong but it is advisable to maintain consistency. Mostly passive is used, but L140 is an example of an active voice. Check this throughout the text.

L134-136: I would suggest stating this as “Results may enhance the biological…and facilitate the development…” If you state it as is, those two terms seem contrasting.

L147: "croaker specimen" - not sure of this term, I suppose it is a mistake.

L176-179: Sentence (explanation) is not clear.

L202: typo, "The" should be "the".

Some additional context would help with L48 in the Abstract. When you mention uniform growth pattern, what do you mean? Uniform among sexes or?

Also, I think there is some clarification needed: throughout the manuscript, authors mention "Leuciscus fish". Is it referring only to the Leuciscus genus? Maybe explain this to make it clearer.

There are a few cases where additional reference would be preferable, L95-96 and L121-124 for instance.

For L95-96 I would suggest not to generalize since the genus is present in Eurasia, and this sentence assumes a much broader area.

In L99-102, if authors are referring to a specific species, then they should mention the species, not just talk about a Leuciscus fish.


The structure of the manuscript is valid, I would only highlight a few points:

L223-226 in the results would be a better fit in M&M section

L335-337 in the discussion is a bit repetitive from the M&M and Results

Regarding tables and figures:

Table 1 - please provide the coordinate system in the description

Table 3 - I think you put the wrong values in the infinite growth column, you put cm in the brackets but I think you listed millimeters (except in the last row 28.97)

Figure 1 - please list the legend in an order as you did in the tables: Female, Male, Total

Figure 2 - please provide a scale for the figure

Figure 5 - on the graph with both sexes, adjust the placement of Z values if possible. The female line is over the male line, but you put the values the other way around, also value for males overlaps with the line for females

Experimental design

The aims and purpose of the research are well-defined.

Methods seem to be appropriate but there are some confusing parts:

L176 I assume you meant to put "t" with subscript 0 instead of "to"

L176-179 this sentence is not clear, nor is the formula itself (L180) since there is no explanation for the variables in the formula.

L181 It is not clear which parameter is this referring to. Growth inflection point or growth characteristics index?

Validity of the findings

Just one observation regarding the validity of the findings.

L296-301: One could argue that the seasonality of sampling makes these comparisons a bit dubious. Coefficient b is often influenced by feeding patterns and reproductive cycles, especially if the sampling is not year-round, so maybe this can be addressed in the discussion. See: https://doi.org/10.1016/j.ejar.2018.11.004
doi: 10.4194/1303-2712-v12_3_17

Additional comments

Dear Authors,

After reading the journal scope and already published manuscripts I think this research is publishable in PeerJ journal. Data is valuable and well-interpreted (with a few limitations that I mentioned in the required fields). I would recommend minor revisions to your manuscript before acceptance for publishing.

·

Basic reporting

The article is written as clear, unambiguous, technically correct text. I understand everything.
The manuscript includes a sufficient introduction that demonstrates how the work fits into the broader field of knowledge.
Relevant prior literature has been appropriately referenced.
The article contains figures and tables, designed at the appropriate level.
The submission is ‘self-contained and represents an appropriate ‘unit of the publication’, and includes all results relevant to the hypothesis.

Experimental design

Original primary research is within the Aims and Scope of the journal.
The research question is well-defined, relevant & meaningful. It is stated how research fills an identified knowledge gap.
The investigation was performed to a high technical & ethical standard in the field.
The authors adhere to the ethical treatment of animals.
Methods are clearly described with sufficient detail & information to replicate.

Validity of the findings

The data are robust, statistically sound, and controlled.
Conclusions are well stated, linked to the original research question & limited to supporting results.

Additional comments

I have some questions and comments.
1) In what size and age did this fish become adult? Why were there not any juveniles during the investigation?
2) Table 1. In my opinion, it needs to add information about the dates of investigation and the number of specimens for every locality.
3) Table 3. I recommended adding current data for Leuciscus chuanchicus.

---

## Round 0.2 · Minor Revisions

Dear Dr. Xiong

I would like to thank you and your co-authors for making the corrections and changes requested by the reviewers. It seems that many scientific problems have been overcome in getting the article accepted, however, the section editor expects you to answer the questions of Reviewer 1. (especially the calculation of asymptotic length and other questions) in more detail in the first version of the review.

Best regards

·

Basic reporting

No comment

Experimental design

No comment

Validity of the findings

No comment

Additional comments

The authors corrected my comments on the manuscript.

---

## Round 0.3 · accepted · Accept

Dear Dr. Xiong

I would like to thank you and your co-authors for making the corrections and changes requested by the reviewers. I read and checked carefully your valuable article and I am happy to inform you that your article has been accepted for publication in PeerJ.

Sincerely yours